# A 12-Year Experimental Design to Test the Recovery of Butterfly Biodiversity in an Urban Ecosystem: Lessons from the Parc Urbain des Papillons

**DOI:** 10.3390/insects14100780

**Published:** 2023-09-24

**Authors:** Magali Deschamps-Cottin, Guillaume Jacek, Louise Seguinel, Clémentine Le Champion, Christine Robles, Mélanie Ternisien, Chloé Duque, Bruno Vila

**Affiliations:** 1Laboratoire Population Environnement Développement, Faculté des Sciences, Campus Saint-Charles, Aix Marseille University, IRD, 3 Place Victor-Hugo, CEDEX 3, 13331 Marseille, France; guillaume.jacek@univ-brest.fr (G.J.); louise.seguinel@gmail.com (L.S.); clementine.lechampion@gmail.com (C.L.C.); christine.robles@univ-amu.fr (C.R.); melanie.ternisien@univ-amu.fr (M.T.); chloe.dqfer@gmail.com (C.D.); bruno.vila@univ-amu.fr (B.V.); 2Laboratoire Géoarchitecture, Territoires, Urbanisation, Biodiversité, Environnement, Université de Bretagne Occidentale CS93837, CEDEX 3, F-29238 Brest, France

**Keywords:** butterfly, biodiversity, experimental park, long-term monitoring, specialist species

## Abstract

**Simple Summary:**

In a world where city dwellers are disconnected from nature, scientists are sounding the alarm regarding the decline in pollinators both in the countryside and in cities. Among these pollinating insects, butterflies are a particular focus of study, as their populations are declining sharply as a result of urbanization and the artificialization of vegetation. An ecological engineering project based on a plantation of host and nectariferous plants and backed up by a well thought out management approach was carried out in Marseille at the Parc Urbain des Papillons (the Butterflies Urban Park). We succeeded in almost doubling the number of butterfly species over the 12 years of the project. Mediterranean species that were rare at the beginning of the monitoring survey colonized the site. However, the monitoring of a nearby natural wasteland shows that some species are still absent. The plant palette used proved its effectiveness, and it would be worth duplicating this system in other urban contexts to encourage butterflies to remain in the city.

**Abstract:**

Urbanization is one of the main threats to biodiversity. However, some urban green spaces could act as refuges for urban fauna if the composition of the flora were less horticultural and if a less intensive management strategy is adopted. Among the taxa, butterflies are experiencing a strong decline from European to regional scales. An ecological engineering project based on a plantation of host and nectariferous plants backed up by a well thought out management strategy was carried out in Marseille at the Parc Urbain des Papillons (the Butterflies Urban Park). We assessed its effectiveness by comparing the butterfly communities in this park before and after the engineering work, and we compared it to a neighboring wasteland with natural habitats. After 12 years of the project, the results show a significant change in the species composition. The species richness greatly increased from 25 to 42 species. Some specialist species we targeted appeared, and their numbers increased from one to five. However, three Mediterranean species are still absent compared to the wasteland with natural habitats. As the plant palette used and the management strategy implemented enabled us to significantly increase the number of species, we now plan to work on the structure of the vegetation.

## 1. Introduction

The loss and alteration of natural habitats caused by land-use changes represent one of the greatest threats to biodiversity [1,2]. Urbanization, identified as a major driver of this loss [3], is characterized, on the one hand, by the significant fragmentation of certain landscapes [4], soil sealing, reduction of green spaces and homogenization of the vegetation composition [5]. On the other hand, urban habitats also host key spaces that represent opportunities for intra-urban biodiversity. The remaining pockets of natural remnants have been shown to be key reserve areas for local biodiversity and rare species conservation [6,7,8]. Urban green spaces, such as parks, gardens and wastelands, represent refuges for urban wildlife and often sustain high species diversity and richness [9,10,11].

Yet, the conflicting demands of social, urban planning and ecological priorities represent a challenge in urban biodiversity conservation [12,13,14]. Small wilderness patches, which are of high conservation value in human-dominated ecosystems such as cities [15,16,17], are often neglected, leading to deliberate or involuntary degradation through redevelopment or pollution, as well as through ecological isolation. Urban parks and other green spaces, which could be seen as potential tools for ecological connectivity between different natural remnants, are mostly composed of non-native ornamental species and are subject to intensive management practices, decreasing the functional capacity of such spaces to support biodiversity. Therefore, local species, mostly consisting of stress-tolerant and generalist species better adapted to urban pressure [18,19,20], are present in small patches of spontaneous vegetation. 

Despite the growing interest in nature-based solutions [21,22], mixed approaches aiming at both the conservation of urban-adapted generalist/ruderal species and local more specialist species in the form of a neocommunity are lacking. This type of approach could represent a real opportunity to enhance urban biodiversity by increasing connectivity between natural remnants and green spaces, creating transient reservoirs for specialist species and increasing habitat quality for more generalist species [23]. Moreover, it would allow for the reconciliation of biodiversity conservation with anthropic interactions in the context of the social use of green spaces in cities.

Among insects, butterflies (Lepidoptera: Rhopalocera) are described as practical indicators for the study of rapid environmental changes and, in particular, the intensity of urban development [24,25]. They are strongly dependent on the composition and availability of vegetation as their feeding source at the adult stage [26] or as host species at the larval stage. At large scale, European butterfly species have shown a strong decline over the last few years, not only in rare species, with 12% of species considered threatened and 31% appearing vulnerable [27] but also in the abundance and distribution of common European species [28,29]. At the city scale, butterfly communities have shown a decrease in the number of species and in the number of individuals spreading from the outskirts towards the city center, with the loss of local butterflies related to the local conditions [18]. It has also been shown that the management practices applied in urban parks or private gardens and the urban matrix accentuate this selectivity of species within the different urban contexts of the city [30,31,32].

On the basis of these results, in 2012, one of the authors, Magali Deschamps-Cottin, set up a unique experimental design system for Mediterranean butterfly research, knowledge sharing and awareness raising located within a former *bastide* estate in Marseille (South-East France): the Parc Urbain des Papillons (Butterflies Urban Park) (BUP). The scientific purpose of this project was to test whether a light engineering approach (plantation and seeding of local Mediterranean species) and management practices adapted to the life cycle of Lepidoptera would enable the return and/or the arrival of new local species and the maintenance of the species present at the site by means of a comparison with butterfly communities of a semi-natural neighboring site.

In this paper, the BUP was assessed as a potential tool for the conservation of butterfly communities by applying engineering methods to conserve and attract butterflies taking into account the social constraints of the urban environment. Firstly, with the creation of the BUP, our aim was to test whether the implemented management and ecological engineering methods, using mixed plantations of local and horticultural species, including host plant species for caterpillars and nectariferous plant for adults, would enhance butterfly biodiversity (richness, abundance and diversity indices) including target species. Secondly, we hypothesized that a mixed approach favors the creation of a neocommunity with an intermediate composition differing from initial conditions and natural remnants but associated with higher diversity due to generalist and specialist co-occurrence. For this purpose, we compared the data sampled at the BUP before the engineering work and for five years after the engineering work and with the neighboring semi-natural site.

## 2. Materials and Methods

### 2.1. Study Sites

**The Butterfly Urban Park (BUP)**—The 1.45 ha site is located in an urban context in the core of the 14th District of Marseille (43°20’14.42” N; 5°24’0.89” E) and corresponds to what was previously part of the 12 ha site of the Domaine de la Tour des Pins, an agricultural wasteland (Figure 1). To the east, the domain is bordered by one of the reservoirs of drinking water of the city of Marseille. To the west, it is surrounded by individual suburban homes with very small surface areas, whereas to the north and the south, it is surrounded by groups of low-rise buildings. In 2010, the management of this site was conferred to the Laboratory Population Environment and Development (LPED) of Aix-Marseille University. The aim was to set up an experimental butterfly conservation site in an urban context and also to provide a tool for scientific training purposes with an educational circuit and training courses for the city’s park managers [33]. The BUP was initially constituted of a mosaic of closed and open habitats dominated by agricultural nitrophilic meadows accompanied by small patches of Mediterranean dry grassland and woodland, orchard and some remnants of Mediterranean shrubland. In 2012, for the purpose of increasing and diversifying the Rhopalocera populations, an ecological engineering project was carried out on half of the nitrophilic meadow. The aim was to neither restore a natural Mediterranean ecosystem nor the butterfly population but to experiment with the implementation of a neo-ecosystem that could then be transferred to other urban parks. The engineering work consisted in the introduction of beds of local host plant species and horticultural nectariferous species (Figure 1). Based on an initial floristic inventory, the park was designed to add host plants for the caterpillars and nectariferous plants for the adults. In all, 15 beds were planted or seeded with 59 plant species. Of variable form, these beds were never wider than 1 m to facilitate observation. Plant selection was based on expert consulting and on an intensive review of the literature on butterfly life cycles and host/nectariferous plant relationships [34,35]. Several butterfly species absent from the BUP were targeted during the development, which led to the introduction of target host and nectariferous plants. A total of 16 host plants were introduced. They include, for example, *Arbutus unedo* for *Charaxes jasius*; *Lotus corniculatus* and *Rosmarinus officinalis* for *Leptotes pirithous*; *Achillea millefolium* for *Melitea didyma*; and *L. corniculatus*, *Medicago lupulina* and *Trifolium* sp. for *Cupido argiades*. Nectariferous plants, such as *Ajuga reptans*, *Aquilegia vulgaris*, *Bupleurum fruticosum*, *Centranthus ruber*, *Cephalaria leucantha*, *Scabiosa columbaria* or *Verbena bonariensis*, were selected for small and large butterflies. Appendix A Appendix A provides a complete list of the selected host and nectariferous plants.

On a larger scale, the BUP is characterized by a varied vegetation structure (open areas with grassland and dry grassland, shrublands, areas with brambles, wooded areas and hedge and border strips) that were conserved because they offer a diversity of habitats accommodating a range of animal species (Figure 1). The park was also designed in such a way as to improve the structure of the site by conserving or developing for the butterflies observation and lookout posts, solariums and refuge areas for heatwaves. In particular, grasslands were maintained open by annual winter mowing (November and December). The aim was also to preserve refuge areas for all stages of development of the butterflies (eggs, caterpillars, chrysalis and imagos) during the winter mowing session. The areas of wasteland and bramble were, therefore, conserved. Annually, at the beginning of spring (April), the planted beds were partially weeded to allow for recovery and to limit competition with the planted species. In order to maintain an attractive area during the summer period (Mediterranean climate), a quarter of the grassland area was watered twice a week for one hour from June to August. This improved area averaged approximately 200 m^2^. 

The Semi-Natural Site—This 0.7 ha site is located 800 m from the BUP (43°20′35.50″ N; 5°24′12.80″ E) on the Colline la Mirabilis. Located on a hill, it is a continuation of the natural environment, which extends this site towards the north. At the foot of the hill, on the south and west sides, there are low buildings. Abandoned more than 70 years ago, it is a postcultural wasteland with *Olea europea* and *Prunus dulcis* that transformed spontaneously into Mediterranean scrubland. The herbaceous layer is mainly composed of *Brachypodium retusum*. The dominant shrub species are *Cistus albidus*, *Rosmarinus officinalis* and *Phillyrea angustifolia*. We also found *Ulex parviflorus*. The tree species present are, in addition to *O. europea* and *P. dulcis*, *Pinus halepensis* and *Quercus ilex*. Mostly composed of open garrigue and considered a semi-natural area, the semi-natural wasteland (WAS) could be a potential species source of butterflies in the BUP (Figure 1). This site was used as a natural site for comparison with the BUP.

### 2.2. Butterfly Field Inventory

The study is based on Rhopalocera surveys carried out annually from 2010 to 2022 in June, July and August. Field inventories were undertaken at the BUP starting in 2010 (except in 2015 because of technical and sampling issues) and at WAS beginning in 2018. At each site, the Rhopalocera Monitoring Plan (RMP) method was followed [38]. This method makes it possible to obtain a relative abundance within a butterfly community along a given path and over a given period. Climatic parameters (wind, temperature, cloud cover and sunshine), which play a crucial role in the daily activity of Lepidoptera [39,40], were also standardized. Temperatures needed to be over 20 °C with a sunshine rate higher than 60% and wind not exceeding 40 km/h [18,38]. For one hour, operators followed a random path, making sure they covered all habitats at the site (meadow, wasteland, pine forest, oak grove, dirt road, orchard, etc.). Then, they captured individuals using a butterfly net. For each capture, individuals were placed in parchment paper wrapping and set aside for the duration of the sampling to avoid counting the same individual several times. At the end of the hour, all butterflies were counted and identified using the Lafranchis butterfly key [41] and then released. Two sampling sessions were conducted each month (June, July and August) at the BUP according to climatic parameters, except in 2010 when only 5 samplings were performed (3 in June and 2 in July). Consequently, a total of 17 sampling sessions were carried out in the BUP_before_, 30 sampling sessions were carried out in the BUP_after_. Over the same 5 year period as BUP_after_ (2018–2022), 42 sampling session were carried out in the WAS, with a minimum of 2 sessions per month.

At the BUP, one of the aims was to attract Mediterranean species of butterflies by reintroducing Mediterranean host species. Therefore, each year we looked for the presence of signs of reproduction on plants, such as egg laying or larvae. In particular, we focused our attention on *C. jasius* L., 1767. This is a specialist species that lay eggs only on *A. unedo* L., 1763, a tree absent in the surrounding environment but planted in the BUP. This means that this butterfly could only be present and reproduce onsite because of the planting in the BUP.

### 2.3. Data Analysis

To compare the species richness, abundance and diversity before and after the ecological engineering project and with the semi-natural wasteland, data sets were selected. Firstly, we used the data corresponding to the 2010–2012 sampling period before planting and constituting BUP_before_. We included the year 2012 considering that the engineering work was too recent to have impacted the butterfly communities. Secondly, we used the data corresponding to the 2018–2022 sampling period after planting (implementation of the engineering work) and constituting BUP_after_. Thirdly, we used the data collected on the Mirabilis semi-natural wasteland (WAS) sampled between 2018 and 2022, as the WAS data were collected only for 5 years.

To better take into account the arrival in small numbers of certain species in the diversity assessment, we used the Shannon diversity index, as it gives greater importance to rare species compared to the Simpson index. Population evenness was assessed using the Pielou index (J), as it is associated to the Shannon index. To characterize the impact of the engineering work on general butterfly population characteristics, we first tested the data normality and homoscedasticity using the Levene and Shapiro tests, respectively, with α = 0.05. As these requirements were achieved for the abundance and species richness, we used one-way ANOVA with α = 0.05 followed by the Tukey HSD test to compare, site by site, the butterfly mean species richness and mean abundance. The statistical difference among the diversity values was assessed using multiple Hutchenson *t*-tests [42], which allowed for the direct comparison of the Shannon and better integrated the index nonlinearity. The Kruskal–Wallis test and associated Dunn test were used to assess the Pielou index, as it did not meet the previously cited requirements for the ANOVA. Those tests were conducted first as a pool (no separation among sampling seasons). The species composition might, however, change during the different sampling periods and butterfly species may not respond in the same way to the engineering work depending on the species composition, climate variables or nutrition sources available. Therefore, in a second step, the data were classed according to the sampling month (June, July and August) to see whether the community response varied depending on the season or time of year. R software (V4.0.3, R Core Team 2020, Vienna, Austria) was used [43]. The Kruskal and Dunn tests were also later used to assess the change in butterfly abundance and proportions over time.

Complementary to the modification of the species richness or abundance, the aim was also to assess changes in the species composition. We, therefore, used multiple nonmetric multidimensional scaling (NMDS) with the vegan R package [44] and Betapart package [45]. Sorensen dissimilarities used in the NMDS function to compare the community composition was based on two phenomena: community nestedness, where differences between two communities are due to species addition, and one community is nested in the other; species turnover. Using the Betapart package from R, we tested these two phenomena to explain the differences among our three communities and represented this using a vegan package in a 2D NMDS. The statistical significance among compositions was then tested using an analysis of similarities (AnoSIM) through a vegan package, which is a nonparametric test similar to ANOVA but operates on a ranked dissimilarity matrix [46].

To assess the species specificity to one or the other conditions and identify potentially lacking key species between the engineered wasteland and natural remnant, we applied the IndVal method [47] using the labdsv package [48]. Species were considered indicators for a specific condition if the *p*-value of the IndVal was below 0.05 and if the *p*-value of IndVal was above 0.7 [49].

## 3. Results

### 3.1. General Characteristics of Butterfly Communities

Over the whole set of sampling sessions (3 years BUP_before_, 5 years PUP_after_ and WAS), a total of 434 individuals of 25 species were recorded in the BUP_before_, 1043 individuals of 42 species in the BUP_after_ and 1708 individuals of 46 species in the WAS (Table 1). Taken together, all sites accumulated a total of 51 butterfly species.

The mean richness observed during the sampling sessions (pooled for June–July–August) varied between 6 and 8 species in the BUP_before_ and 12 and 7 in the WAS. The species richness was significantly lower in the BUP_before_ compared to the two other conditions but only for the month of June (Table 1). 

Mean diversity varied between 1.4 in the BUP_before_ in August and 2.4 in the WAS in June. As observed for the species richness, the species diversity was only significantly lower in the BUP_before_ in June (Table 1). 

The minimum mean abundance was observed in August in the BUP_after_ with 21.2 individuals, and the maximum mean abundance was observed in June in the WAS with 56.2 individuals (Table 1). The abundance in the BUP_before_ was significantly lower than the abundance in the WAS; yet, no significant difference was observed between BUP_after_ and WAS, as well as BUP_before_ (Table 1). The individual abundance significantly decreased from June to August at all sites (BUP_before_: *p* = 0.0069, *t* = −3.22; BUP_after_: *p* = 0.031, *t* = −2.59; WAS: *p* = 1.2 × 10^−4^, *t* = −4.71).

The species equity calculated using the Pielou index was significantly lower in June in the BUP_before_ (Table 1). No significant difference could be observed in the pooled data.

### 3.2. Impact of the Planted Bed on Targeted Butterfly Species

A total of 19 new species appeared in the BUP after the engineering project and two species disappeared (Figure 2). The new species and associated planted/seeded host plant larvae were the following (based on [41]): *C. jasius–A. unedo*; *Coenonympha pamphilus–Poaceae family*; *C. argiades–Trifolium/Fabaceae*; *Cyaniris semiargus–Fabaceae* (*particularly T. pratense*); *Euchloe crameri–Brassicaceae*; *Gonepteryx cleopatra–Rhamnus* sp.; *Gonepteryx rhamni–Rhamnus* sp.; *L. pirithous–Fabaceae & Salicaria* sp.; *Limenitis reducta–Lonicera* sp.; *M. didyma–Scrophulariaceae & Plantago* sp.; *Muschampsia baeticus–Marrubium* sp. *& Ballotus* sp.; *Muschampsia flocciferus–Stachys* sp.; *Pieris mannii–Brassicaceae*; *Pontia daplidice–Reseda* sp. *& Brassicaceae*; *Pyrgus armoricanus–Potentia* sp.; *Pyrgus malvoides–small Rosaceae*; *Satyrium esculi–Quercus ilex & coccifera*; *Thymelicus sylvestris–Poaceae*; and *Vanessa cardui*–multiple plant family (Figure 2b). The two species that disappeared are *Libythea celtis*, whose caterpillar feeds on *Celtis australis*, and *Plebejus argus,* of which the caterpillar is often found on plants from the *Fabaceae* family. All species present in the BUP after ecological engineering were also present in the WAS, except four (Figure 2): *Cacyreus marshallii*, *C. argiades*, *P. armoricanus* and *P.* Some species were only observed in the WAS: *Aglais urticae*, *Anthocharis cardamines*, *Fabriciana niobe*, *Hipparchia fidia* and *Melanargia occitanica.*

Among these new species at PUP_after_, five are Mediterranean species: *G. cleopatra*, *L. pirithous*, *L. reducta*, *M. baeticus*, *P. mannii* and *C. jasius*. *C. jasius*, our target species, arrived 7 years after engineering started. Since it was established, breeding has been observed every year.

### 3.3. Patterns of Change in BUP Butterfly Community Composition

#### 3.3.1. Changes in the Community Composition over Time

With at least 50 individuals captured per year, the following species were the most frequently recorded species over the 12 years of monitoring: *Aricia agestis*, *Brintesia circe*, *Carcharodus alceae*, *Lasiommata megera*, *Maniola jurtina*, *Pieris napi*, *Pieris rapae*, *Polyommatus icarus*, *Pararge aegeria* and *Lycanea phlaeas*. *M. jurtina* was the most abundant species with 437 individuals captured, all sites included (Figure 3a). *P. rapae* (16,1%) and *M. jurtina* (40.6%) were the dominant species in the BUP_before_, corresponding to 56.7% of all captured individuals (Figure 3c). After the ecological engineering work, the contribution of *M. jurtina* to the total species composition significantly dropped from 40.6% (2010–2012 period) to 13.3% (2022), whereas the number of individuals captured each year stayed constant, as shown by the steadily increasing accumulated number curve, as shown, respectively, in Figure 3a,c. P. *napi*’*s* contribution, which represented 6% of the community before the ecological engineering work, significantly dropped to 0.8% after (Figure 3c), with only a few individuals observed since 2018 (Figure 3b).

In contrast, some species such as *C. alceae*, almost absent before the ecological engineering work, were recorded (0.4% in 2010, none in 2012) and now present a steady population (between 2.4 and 3% for 2018 and 2021 and 8% in 2022, Figure 3). Other species presented a significant increase in individuals and total contribution. This is the case for *P. icarus* of which 7 individuals were captured between 2010 and 2012 (2.3 per year on average) and 126 individuals between 2018 and 2022 (25.2 per year on average) and *P. aegeria* with 2 individuals in the 2010–2012 period (0.6 per year on average; 0.54% overall contribution) and 42 over the period 2018–2022 (8.4 per year on average; 7.9% overall contribution) (Figure 3a). 

The number of individuals of *L. megera* also increased from 7 per year on average in the BUP_before_ to 14 per year on average (70 in total) in the BUP_after_, but its contribution to the total community did not change significantly (Figure 3b).

The less common species, here defined as other species, also significantly increased in number (Figure 3a) but not in terms of contribution (Figure 3c). Neither a significant change in contribution nor captured number were observed for *B. circe*, *P. rapae*, *L. phlaeas* or *A. agestis*. The engineering work induced a delay of several years depending on the species considered. The most changes in the rates of individuals captured were observed in the 2016–2018 period (Figure 3a,b). This delay (?) was also observed for the targeted Mediterranean species such as *C. jasius*, which took 7 years to be observed and to reproduce in the BUP.

#### 3.3.2. Trajectory of the Community Composition 

In order to assess the trajectory of the butterfly communities, we compared the community composition based on the presence/absence data among the BUP_before_, BUP_after_ and WAS. Butterfly communities from the BUP_before_, BUP_after_ and WAS significantly differed in terms of composition (Figure 4a). In the NMDS, the communities in the BUP_after_ were intermediate between those of the BUP_before_ and WAS (Figure 4a). Most butterfly species present in the BUP_before_ were also present in the BUP_after_ and WAS, as shown by the nestedness NMDS (Figure 4b). Therefore, differences observed between the BUP_before_ and the two other conditions are, to a significant extent, due to a species addition. This was also confirmed by the NMDS performed on the community turnover that was not significant in the BUP_before_ (Figure 4c). However, the significant differences observed between the BUP_after_ and WAS do not seem to be because of species addition (Figure 4b). Even if species addition is possible, it seems that this was not the major cause of the differentiating communities (Figure 4b).

The differences in community composition were mainly due to species turnover meaning that BUP_after_ and WAS communities have their own species. To identify which were the key species leading to these differences, an IndVal test was conducted (Table 2). Only four species were characterized with a *p*-value < 0.05 and an IndVal higher than or equal to 0.7 (e.g., [49]). *M. occitanica*, *E. crameri* and *P. daplidice* seem to present a high degree of specificity in the WAS, whereas *P. napi* present a high specificity in the BUP_before_. No species seems to present a high specificity in the BUP_after_.

## 4. Discussion

### 4.1. Impact of Species Phenology on Results

The diversity both in the WAS and BUP_after_ was significantly different from the BUP_before_ for the month of June. This result is probably linked to the constraints observed under the Mediterranean climate, notably in terms of precipitation and temperature, which lead most butterfly species to have their phenological peak from May to early July [50,51]. This period is then followed by a decrease both in abundance and specific richness of the communities, as observed in [52]. This peak in butterfly diversity also coincided with food availability due to the Mediterranean vegetation phenology [53,54,55,56]. The WAS and BUP_after_ were mostly composed of early blooming Mediterranean species or nectariferous horticultural species that are very sensitive to summer drought leading to a drop in nectar availability after May. Conversely, the BUP_before_ represented a mesophilic wasteland dominated by *Holcus lanatus*, with a few nectariferous species inducing less seasonal variation.

### 4.2. Ecological Engineering: Effects of Host and Nectar Source Diversification on Butterfly Community

Over a 12-year period, we show that the diversification of nectar sources and selected host plants in the BUP led to a significant increase in butterfly species richness, close to the level observed in the natural remnant (WAS). As most butterfly species are, to varying degrees, specialists of one plant species, genus or family, the availability and quality of nectar produced by the plant communities and the host plant, both present and introduced, are considered key factors in the composition of butterfly communities and their fitness [28,51,52,53,54]. For example, the increase in the proportion of *P. icarus* could be explained by the increased abundance of plants from the Fabaceae family that were seeded or planted (e.g., *Lotus* sp., *Medicago* sp. and *Trifolium* sp.) in comparison with the BUP_before_, which was dominated by Poaceae. The same explanation applies to *E. crameri* and *P. daplidice*, which colonized the BUP_after_ following the introduction of their host plants of the *Brassicaceae* family (*Biscutella laevigata* or *Arabis hirsuta*) or *M. didyma* with the introduction of *A. millefolium*.

Therefore, no significant difference could be found between the BUP_before_ and the BUP_after_ regarding the global butterfly abundance. This could be explained by multiple factors. For example, it has been shown that species preferentially observed in natural environments are generally present in lower abundance [57]. On the other hand, the number of individuals of some species could have been limited by the small surface areas (200 m^2^) concerned with by the engineering work.

Finally, the proportion of species over time showed a seven-fold delay in responding to the BUP engineering changes, as observed in terms of the richness and abundance by Waltz and Wallace Covington [58]. Observing this delay, we might also emphasize that with more time, the abundance observed in the BUP_after_ could become significantly higher than in the BUP_before_, as it is already nonstatistically different from the natural remnant. Further monitoring should be conducted to see whether the butterfly abundance continues to increase or whether it has reached its limit due to the presence of other limiting factors previously cited, such as the park size and host species abundance.

### 4.3. Ecological Engineering: Microhabitat Alterations to the Butterfly Community

While we did not observe any significant variation in the overall abundance between BUP_before_ and BUP_after_, it was possible to observe a shift in the abundance of the generalist species. Thus, the contribution of very common generalist species, such as *M. jurtina* and *P. napi*, relative to the total species composition dropped significantly from BUP_before_ to BUP_after_. As host plants of these species are always observed at the site, this is probably not the factor at play. Most butterflies require a habitat with a diversity of space adapted to all stages of development, including host plants for larval development, availability of litter, sources of nectar for adult and an area for warming [59,60,61]. These habitats are characterized by the vegetation and the structure of their strata [60]. As underscored by multiple authors, the degradation and management of the structure of the vegetation strongly impact butterfly communities [25,61,62,63,64], even in urban parks [60,65], where it appears that management practices and design influence habitat characteristics. It can be assumed that the management carried out in the BUP creating a mosaic of habitats, such as brambles, areas of high grassland, afforestation, close-cropped lawns and mixed plantations, could have played a key role in the shifts in butterfly diversity and composition. This might have improved the habitat resources for smaller species, such as *C. alceae* and *P. icarus*. In contrast, *M. jurtina* caterpillars feed on high and broad-leaved grasses (e.g., *Brachypodium* sp., *Festuca* sp. and *Poa* sp.); *P. napi* mostly feeds on *Alliaria officinale*; and *Cardamine pratensis* and *L. phlaeas* mostly feed on *Rumex* species [41]. All these species were common on the mesophilic grassland present in the BUP_before_. The absence of an increase in the numbers of *Maniola jurtina* or *Lycanea phlaeas* compared to almost all other species and the decrease in *Pieris napi* could be due to the decrease in the mesophilic grasslands and clearings [65].

### 4.4. Role of the BUP in Mediterranean Butterfly Conservation: Appearance and Reproduction of Targeted Species 

With the creation of the BUP, we targeted several butterfly species, in particular Mediterranean species. The impact of the ecological engineering work was tested using an indicator species, *C. jasius* L., which was absent from the BUP_before_. Its caterpillars were detected for the first time in 2020, which means 8 years after the start of the engineering work, whereas *C. jasius* is a species with large wings, testifying to a high dispersion capability. Since 2020, it has been observed each year. These results raise the question of the detection of the host plant by butterflies. Planted at the beginning, *A. unedo* was probably too small to be detected or considered as a sufficient resource to feed caterpillars by the butterfly. For smaller species, we may also question their capacity for dispersion to (re)colonize spaces where they are absent. This implies that applying engineering methods to conserve and attract butterflies requires consideration at the local scale (detection and sufficient host plants) and at the landscape scale to take into account the dispersion capacity of species to colonize the site even though the colonization of the BUP by small species (*L. pirithous*, *S. esculi*, etc.) suggests that the process depends more on their detection and presence in a sufficient quantity of the host plant. Finally, the intensification of urbanization observed around the BUP also raises the question of attracting local species in a place that could be disconnected from other biodiversity reservoirs in the near future [59,66].

### 4.5. Toward a Specific Community: The Theory of the Neo-Ecosystem

Although the BUP_after_ and WAS had a similar specific richness, they differed in community composition. Some typical Mediterranean species were found in the BUP_after_ (*G. cleopatra*, *L. pirithous*, *L. celtis*, *M. baeticus*, *P. mannii*, *Pyronia bathseba*, *Pyronia cecilia* and *S. esculi*) but were absent in the WAS. Inversely, some Mediterranean species present in the WAS (*H. fidia*, *Coenonympha dorus* and *Satyrium ilicis*) were absent in the BUP_after_. These typical species of dry grasslands, shrubland and stony moors [41] probably do not find this type of habitat in sufficient proportion in the BUP_after_. On the other hand, BUP_after_ presents generalist butterfly species that are common and abundant within urban spaces (*P. rapae*, *P. aegeria*, etc.) [31]. It follows that the BUP_after_ represents a space at the interface between urban green spaces, such as the BUP_before_, and a natural environment, such as the WAS. This type of intermediate environment can be assimilated into the concept of a neo-ecosystem, or emerging ecosystems that “result when species occur in combinations and relative abundances that have not occurred previously within a given biome” [67] with, at a smaller scale, the formation of neo-communities.

## 5. Conclusions

This 12-year experiment demonstrates that while the abundance has not changed, the species composition has strongly been transformed. The specific richness has increased sharply, from 25 to 42 species, and several Mediterranean species, which we targeted, have appeared. This shows that with thoughtful planning and a good management strategy, it is possible simply and fairly quickly to reverse the trend of biodiversity erosion.

Therefore, even if many of the species, rare or absent in Marseille, have been observed at the BUP_after_ in connection with the habitat diversity found there (for example, *Satyrium w album* in woods and cool edges, *C. semiargus* in mesophilic to humid meadows or even *M. flocciferus* in various meadows with the presence of its host plant), with the management applied, our results also show the presence of species specific to the WAS, such as *P. daplidice*, *E. crameri* (observed in large numbers) and *M. occitanica*. These butterflies are characteristic of open and dry environments, such as Mediterranean grasslands or shrubland [41], observed at the WAS but which are difficult to recreate in the BUP. This suggests that the presence of host and nectariferous plants is insufficient and the structure of the habitat is important (vegetation strata, extent of the habitat patch and disposition of the habitat) (e.g., [68,69]).

Finally, from a practical point of view, it is now necessary to test the replication of this experimental design. This was the case in 2022, in the municipality of Aix-en-Provence near Marseille, where the BUP has been replicated. The replication is situated in an urban park, within the periphery of the city center. Butterfly surveys were conducted prior to its establishment, and it is expected that in future years the impact of the engineering work will be documented. 

## Figures and Tables

**Figure 1 insects-14-00780-f001:**
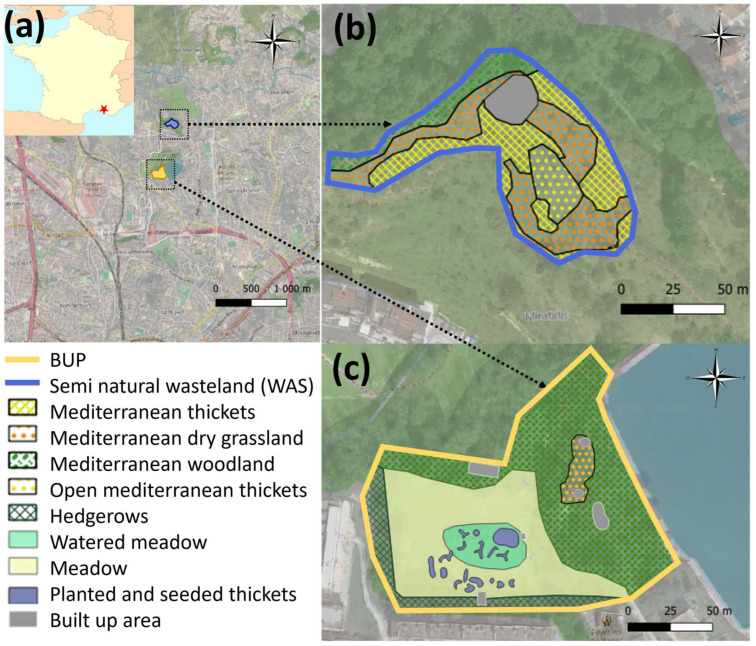
Locations and habitat boundaries of the studied sites: (**a**) Red star: city of Marseille; 14th District of Marseille, France; (**b**) habitats present at the semi-natural site, Mirabilis wasteland (WAS); (**c**) habitats present in the butterfly urban park (BUP) after the ecological engineering work. Map font source: (**a**) Wikicommons, OpenstreetMap2022 [36]; (**b**) Google 2022 [37].

**Figure 2 insects-14-00780-f002:**
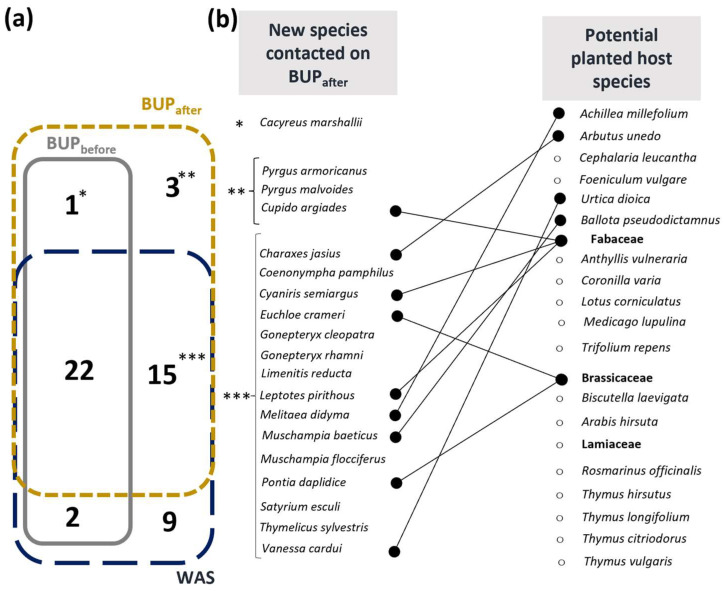
Newly observed species in the BUP after engineering work and potential planted host species: (**a**) Venn diagram showing the number of species common to or unique to each tested condition; (**b**) list of newly appeared species in the BUP after engineering work and potential associated planted or seeded species.

**Figure 3 insects-14-00780-f003:**
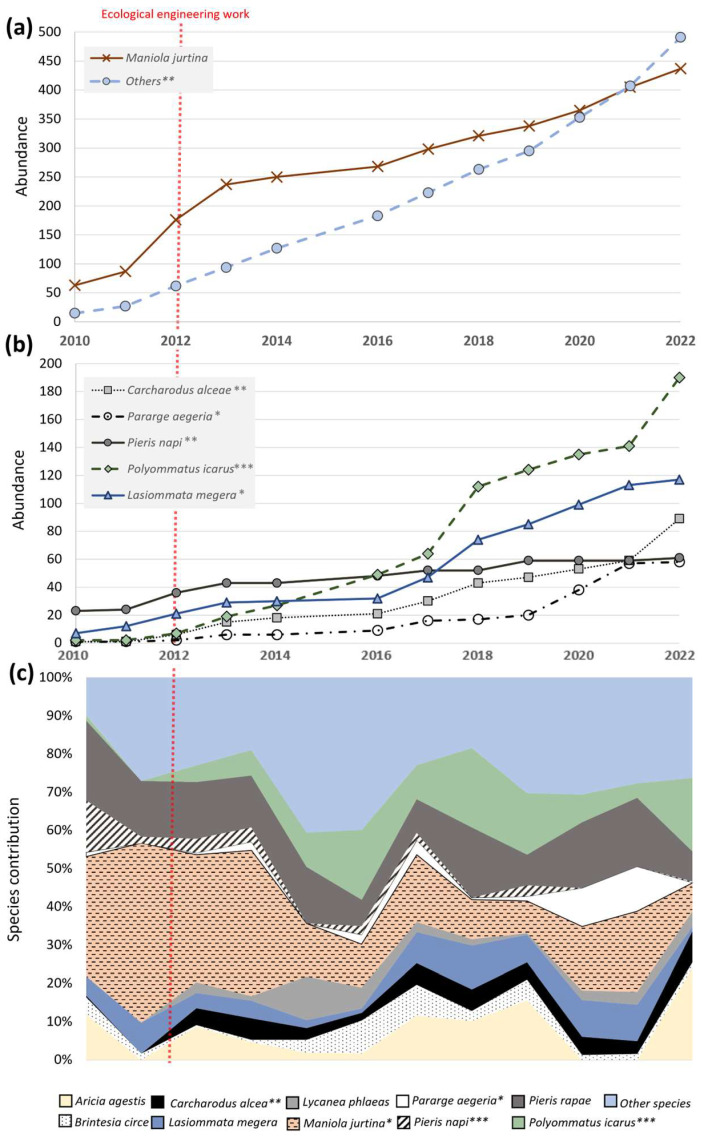
Changes in the number of individuals and contribution of the 10 most contacted butterfly species over the 12 years of inventories. (**a**) Accumulated abundance of the most abundant species, with *M. jurtina* compared to the other species. (**b**) Accumulated abundance of the 5 most abundant species, without *M. jurtina*. For (**a**,**b**), an increase in the slope of the curve indicates an increase in the number of individuals caught in year n compared with n−1. Conversely, if the slope of the curve decreases, the number of individuals caught in year n decreases compared with n−1. (**c**) Contribution of the 10 most abundant species as a percentage of the total butterfly population. * Means that the differences between BUP_before_ and BUP_after_ are statistically significant (Kruskal–Wallis test: * *p* ≤ 0.05, ** *p* ≤ 0.01, *** *p* ≤ 0.001). The red, dotted, and vertical line represents the year in which the ecological engineering work was conducted.

**Figure 4 insects-14-00780-f004:**
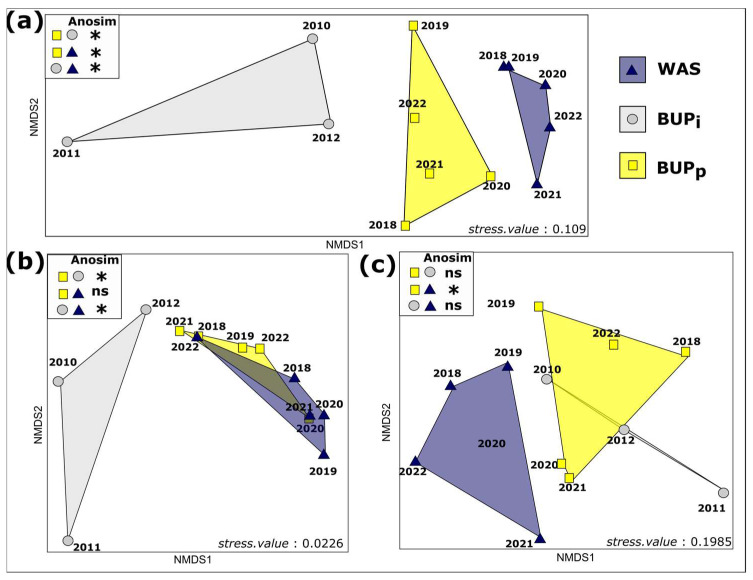
Nonmetric multidimensional scaling of butterfly communities observed in the BUP_before_ and BUP_after_ and WAS: (**a**) NMDS of butterfly communities observed in the BUP_before_ (grey circle), BUP_after_ (yellow square) and WAS (blue triangle) based on presence/absence data and Sorensen–Dice index; (**b**) NMDS based on community nestedness, in which if communities are overlapping, their community nestedness is not validated and one community is not nested in the other; (**c**) NMDS based on community turnover, whereby if communities do not overlap, there is a significant difference between them due to species turnover. Differences in the composition between the communities were tested using an analysis of similarity (AnoSIM) based on a distance matrix (* *p* ≤ 0.05, ns: no significant difference).

**Table 1 insects-14-00780-t001:** Butterfly richness, mean richness, mean abundance, mean diversity and Pielou index for BUP_before_, BUP_after_ and WAS in June, July, August and pooled.

		BUP_before_	BUP_after_	WAS	*p*-Value ^1^
Total species richness		25	42	46	
Mean species richness	June	6.00 ± 3.38 ^a^	15.30 ± 3.83 ^b^	16.90 ± 3.64 ^b^	***
	July	8.60 ± 2.97	11.70 ± 3.53	11.50 ± 4.60	ns
	August	6.25 ± 3.40	8.30 ± 2.67	9.90 ± 2.91	ns
	Pooled	6.82 ± 2.68 ^a^	11.77 ± 3.68 ^b^	12.78 ± 3.88 ^b^	***
Mean abundance	June	33.75 ± 22.47	50.50 ± 26.10	56.20 ± 13.80	ns ^2^
	July	20.80 ± 11.12 ^a^	32.70 ± 12.32 ^ab^	38.60 ± 14.79 ^b^	*
	August	12.00 ± 5.42 ^a^	21.20 ± 12.05 ^ab^	29.90 ± 14.31 ^b^	*
	Pooled	24.82 ± 15.79 ^a^	34.80 ± 13.92 ^ab^	41.66 ± 14.37 ^b^	*
Mean diversity (Shannon index)	June	1.15 ± 0.63 ^a^	2.34 ± 0.28 ^b^	2.44 ± 0.30 ^b^	***
	July	1.86 ± 0.30	2.12 ± 0.29	1.91 ± 0.56	ns
	August	1.45 ± 0.74	1.78 ± 0.34	1.92 ± 0.39	ns
	Pooled	1.43 ± 0.51 ^a^	2.08 ± 0.29 ^ab^	2.08 ± 0.42 ^b^	*
Pielou index	June	0.61 ± 0.28 ^a^	0.87 ± 0.03 ^b^	0.87 ± 0.06 ^b^	*
	July	0.89 ± 0.06	0.88 ± 0.06	0.79 ± 0.13	ns
	August	0.84 ± 0.11	0.88 ± 0.12	0.85 ± 0.11	ns
	Pooled	0.75 ± 0.17	0.87 ± 0.06	0.83 ± 0.09	ns

^1^ Depending on the data type and requirement (normality, homoscedasticity, independence), different statistical tests were used: ANOVA and Tukey HSD were used for richness and abundance; multiple Hutcheson *t*-test for comparing the Shannon index, as it better takes the nonlinearity of the Shannon index into account; and the Kruskall–Wallis test followed by the Dunn test for the Pielou index, as it did not meet the parametric test requirements. *** *p* ≤ 0.0001, * *p* ≤ 0.05, ns: no significant difference. Different letters mean that the samples were statistically different from each other. ^2^ ANOVA test was nonsignificant; it should be pointed out, however, that the *p*-value was 0.06 and that the *t*-test conducted in the exploratory part of the research did show a significant difference between BUP_before_ and WAS with a *p*-value of 0.03 and *t* = −2.55.

**Table 2 insects-14-00780-t002:** Indicator value (IndVal) and relative frequency of each species at the different sites. IndVal represents the specificity of a species for one of the three tested conditions. Relative species frequency corresponds to the proportion of the sample where the species was present. *p*-Value corresponds to the statistical significance of the IndVal. All IndVal are available in Appendix A Appendix A. * *p* ≤ 0.05, ** *p* ≤ 0.01.

Species	IndVal			Relative Species Frequency			*p*-Value
	BUP_before_	BUP_after_	WAS	BUP_before_	BUP_after_	WAS	
	_(2010–2012)_	_(2018–2022)_	_(2018–2022)_	_(2010–2012)_	_(2018–2022)_	_(2018–2022)_	
*Melanargia occitanica*			0.800			0.8	*
*Euchloe crameri*		0.033	0.833		0.2	1	**
*Pontia daplidice*		0.114	0.714		0.4	1	*
*Pieris napi*	0.714	0.114		1	0.4		*
*Gonepteryx cleopatra*		0.355	0.556		0.8	1	*
*Melitaea didyma*		0.356	0.556		0.8	1	*
*Vanessa cardui*		0.625	0.225		1	0.6	*
*Pyronia bathseba*	0.072	0.026	0.652	0.3	0.2	1	*
*Pyronia cecilia*	0.048	0.429	0.429	0.3	1	1	*
*Thymelicus acteon*	0.048	0.429	0.429	0.3	1	1	*

## Data Availability

Supplementary information or data provided upon request.

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
