# Peer review of "A 12-Year Experimental Design to Test the Recovery of Butterfly Biodiversity in an Urban Ecosystem: Lessons from the Parc Urbain des Papillons"

_insects, 2023, doi:10.3390/insects14100780_

Round 1

Reviewer 1 Report

The paper

I would like to thank you for the opportunity to prepare the review for the paper titled A 12-year experimental design to test the recovery of butterfly bio- 3 diversity in urban ecosystem: lessons from the Parc Urbain des Papillons.

The manuscript is well done and almost complete although I suggest trying to add some more recent literature to compare the data, especially with Mediterranean countries. The key concept is well formulated. The research topic in that formulation is very interesting and not much developed in other previous reviews or articles.

The authors were quite rigorous in covering all the adequate breadth & depth.

The manuscript is well done and complete although I suggest trying to add some more recent literature to compare the data. In addition, the authors should report other data especially from Mediterranean countries.

The key concept is well formulated. The research topic in that formulation is very interesting and not much developed in other previous articles. The authors were quite rigorous enough to cover the adequate breadth & depth.

Unfortunately, I need to present some critical remarks. An aspect that does not convince is that the study is based on Rhopalocera surveys carried out annually from 2010 to 2022 (line 165)  in June, July and August.  The chosen sampling period is very limited especially considering that the flight pick of butterflies is generally in the full spring times and not in summer. Moreover, species can be monovoltine, bivoltine or plurivoltine and a limited period of sampling can just not detect the species because the flight period does not fall in time of the sampling survey.  Sampling also in April and May would be more profitable to see the real Lepidopteran fauna. Please consider explaining better the scientific reason of your choice. Climate conditions also influenced the butterflies' composition over time. It is a pity not to have the butterflies community in WAS before BUP. Could you have some previous data at least on the presence of species in WAS before the experiment or at least at the beginning of it?

Please consider comparing the flight period of butterflies with more extensive literature e.g.: in Manachini B, Bazan G, Schicchi R. Potential impact of genetically modified Lepidoptera-resistant Brassica napus in biodiversity hotspots: Sicily as a theoretical model. Insect Sci. 2018 Aug;25(4):562- 580. doi: 10.1111/1744-7917.12588. Epub 2018 Apr 26. PMID: 29536624.  You can find an updated survey of the flight period of several butterfly’s Mediterranean species. 

Another important aspect that is not mentioned is if pesticides, especially insecticides were applied inside the areas or in the surrounding areas. This aspect is very important because can influence the recruitment of some species that have different susceptibility. In 12 years, several laws were changed in terms of active compounds and control methods that could be used in urban areas. For example, products  used against mosquitos (see for example permethrin)  can have adverse effects on butterflies (see or example: 1. Mulé R, Sabella G, Robba L and Manachini B (2017) Systematic Review of the Effects of Chemical Insecticides on Four Common Butterfly Families. Frontiers in Environmental Science Front. Environ. Sci. 5:32. doi: 10.3389/fenvs.2017.00032. and 2.    Hoang, T. C., Pryor, R. L., Rand, G. M., and Frakes, R. A. (2011). Use of butterflies as nontarget insect test species and the acute toxicity and hazard of mosquito control insecticides. Environ. Toxicol. Chem. 30, 997–1005. doi: 10.1002/etc.462)

Another remark is to add the name of the scientist and the order and family of the species e.g. at line 130 Arbutus unedo for Charaxes jasius;  should be: Arbutus unedo L. (Ericales: Ericaceae) for Charaxes jasius; L. (Lepidoptera: Nymphalidae). This should be done for all specie both animals and plants.

Moreover, remember that when a specific scientific name is mentioned more than 1 time successive times the genus should be pointed e.g. Carcharodus alcea at line 299 will be C. alcea when mentioned at line 320.

SPECIFIC REMARQUES

Line 193-195 Please specify you report the data of WAS only for a period of 4 years (from 2018-2022) and not for all 12 years.

Line 202 - 203. Please explain better why you use univariate analyses to compare diversity indices. Normally no statistical analysis is done on indices. In fact, most diversity indexes behave non-linearly, while most statistical tests assume linearity. So, no, you cannot just use ANOVA and t-tests to compare them. Please explain better your approach.

Line 269 and over. The authors claimed new associations between specific plants and specific butterflies, however, they do not mention for each species if the association was as feeding plants for adults or they are host plant larvae. In addition, many of these associations are already reported in the literature.

Line 449 do you mean Pieris napi or Brassica napus? Please correct the name.

Please check puntation. 

Line 449 do you mean Pieris napi or Brassica napus? Please correct the name.

Reviewer 2 Report

 As public awareness of butterfly/insects declines attain momentum, there are, across Europe and elsewhere, increasing numbers of small local projects attempting to ameliorate the situation with involvement of general public, sometimes represented by municipalities. The butterfly gardening, insect-friendly management of urban parks, etc., are much welcomed initiatives, suffering the shared problem that the outcomes are only rarely academically evaluated and publicised. Hence, I much welcomed this manuscript, which is the more valuable that it targets a location in Mediterranean Europe, i.e., within a global biodiversity hot-spot with specific threats, but also large numbers of potentially endangered endemics – and in a region, where all the “help the butterflies” initiatives seem to be less developed, than, say, in NW Europe.

Yet another positive side of the manuscript it that it is based on reasonably long-term, and systematically collected data.

Unfortunatelly, however, the study has multiple serious flaws, some of them ameliorable by more careful writing/editing, some, I am afraid, originating in the stage of data collection and hardly ameliorable. 

To start with, it is based on comparison of only two sites – and three situation – one being the eco-engineered city park, the other being a “natural vegetation” nearby. Here, although I fully understand that replicates of the “engineered” park were not available, and single-site case studies are fully warranted, I wonder why not more “natural vegetation” replicates? It would give way much stronger support for the findings of the study. Unfortunately, I understand that the missing data can hardly be collected now, but what about having at least a single-season data from multiple natural sites?

Given the above, the authors should be extremely aware of this shortcoming in both analyses, and their interpretation. What I would suggest, is not to build the paper on the comparison of the two sites (because the natural site is just “a” natural site, not a representation of such sites in the environs of the intervention site. Instead, I would build it on a analysing the temporal changes within the intervention site, perhaps using an ordination analysis, focused on species, with time, in years, or with pre- and post- intervention situations, as predictors. The NMDS analysis is a right step in this direction.

Similarly, I would avoid too much details into species that “doubled” or not (most of the text at lines 320-339), as these things are always tricky, given the variation in weather, activity of butterflies, etc. The suggested ordination analysis would deal with this.

Third, I have problem with splitting the analyses by months (which are only artificial divisions of time!), and perhaps bigger, but not redeemable, problem with the temporal scope. Specifically, why you omitted May, which is important moth in Mediterranean phenology, and included August? Well, Charaxes jasius is the August species, so August is OK, but omission of May should be explained/appologised.  In any case, all the separate tests by months are statistical and biological nonsense, esp. as tests for pooled data are missing. Get rid of the by-months analyses, they are only misleading, and work with pooled data.

Fourth, the description of statistical methos (lines 202-205) is superficial, to state it politely. You should not write “Differences between butterfly community characteristics SUCH AS (emphasis mine) species richness, 202 abundance or Shannon diversity index…” – you should exactly list, what has been done, including specifications, why were these characteristics targeted. Why Shannon and not, e.g., Simpson, and why is the Pielou not mentioned?

Five, at lines 105 and below (Description of the study sites), descrining the environs of the two sites is missing. We know what city arrondissement it is, but how it looks like there? Tall or low buildings, gardens or closed blocks, etc. From the map (Figure 1) I see some green spaces, what is their character?

The description of the natural site is even more superficial. "Short garrique" is a nice characterisation of vegetation, but what about dominant plants, at least? Again, no information on the surrounding is given. Is the site managed somehow?

Finally, I have a reservation about the sources cited at the beginning of Introduction, while justifying the project of butterfly conservation in urban settings. In my opinion, you missed some earlier published studies targeting specifically butterflies – I may self-servingly pinpoint one (Jarosik, V, et al. 2011, Biological Conservation 144, 490-499), but there exist many others, and the decision is yours.

In summary, 

the paper has potential, but should be considerably re-analysed, perhaps shortened, turned, in essence, into a "Short note" format, highlighting the strong findings and supressing the weak points. 

Minor remarks

136 vegetal structure - change to vegetation structure, as "vegetal" is French, but not English, word (they are similar, I know)

197-200: move this to the previous section (2.2), which describes the field procedures. 

Caption to Figure 1 - I would change "habitats observed" to "habitats present"

Line 320: Carcharodus alcea -> Carcharodus alceae

Line 337: alsoobserved => also observed

Line 412: orMelitaea didyma -> or Melitaea didyma

Line 438: „As has been recently underscored by Han et al. [52] on an urban park…“ In the list of references, Han et al. are number [53]. Now, I understand that the numbering system is difficult for authors, but you must thoroughly check it before resubmission.

Line 444: You should not write that some larvae “appreciate” something in scientific text. Consume, feed on, develop on… are better options.

Line 450: A geographically closer reference, i.e. st. from Europe, should illustrate the decline of common species in managed sites, next to that from China. I might supply one of mine own, but as it is against the policy of the Journal, you should search it yourself.

Line 492: “Evolved” -> perhaps “Transformed”, as “evolved” implies biotic evolution, which is much longer process.

Line 472: “If BUPafter and WAS had…” -> “Althoug BUPafter and WAS had…”

Lines 504-6. This statement, although correct, looks as your discovery if presented without accompanying citations. Which is, of course, false, many authors from Europe, and beyond, have discussed this crucial aspect. Search for papers and books by Roger Dennis, Tim Shreeve, Jeremy Thomas, and many others.

Figure 2 – The French word “autre” should not be in the graph of English-written text. Should be “others”?

Not much to comment above, it needs thorough care for the languge during revision.

Reviewer 3 Report

Dear Authors,

I have no comments regarding your research. You have made a good manuscript on the materials of long-term observations, which will be of interest to a wide range of entomologists. However, I would recommend some corrections regarding the formatting of the manuscript.

- In my opinion, there is no point in italicising French names in the text.

- Some of the French names may be replaced with English, as not all readers know French, may be unclear: 1) bastide – town, 2) arrondissement – district. Of course, I'm not a good expert in French, I can be wrong.

- Line 84. the Parc Urbain des Papillons ie. the Butterflies Urban Park. It should be i.e., and without italic.

-  An English translation of the name of this place (the Butterflies Urban Park) may be added to both summary and abstract.

- Line 68. (Lepidoptera rhopalocera). It should be without italic. Correct with a colon, R – capital letter. Lepidoptera: Rhopalocera.

- Lines 130 and 183, etc. You need to give the full name of the species at the first mention in the text. Please check this throughout the text, this error occurs elsewhere in the manuscript.

- Line 132. Trifolium sp for. “sp.” should be without italics and with a full stop. As in the previous case, please check this throughout the text, this error occurs elsewhere in the manuscript.

- Lines 109, 139, 281, etc. – Fig. 1, Fig1, Fig 2. The formatting should be uniform, according to the requirements of the journal.

- Lines 231, 232, 237, etc. In a number of places there are italics, in others there are not. Should be checked in the text of the manuscript.

- Figure 3. In my opinion, the figure doesn’t look good. I would recommend improving them, especially the third (c). Here it would obviously be good to make it coloured. In the second (b) the dots are merging into each other; clarity would need to be improved. (a) “autre” is the French word, it should be “others”.

- Lines 483 and 484. In my opinion, there is no point in italicising the quote.

- Supplementary materials. The formatting of plant names should be checked. “sp.” should be without italic, Aster nova angliae should be Aster nova-angliae.

Round 2

Reviewer 1 Report

Dear authors,

thank you for improving the manuscript according to the suggestions.

Author Response

Dear colleague

Thank you for your comments, which have improved this article.
I hope that this last version will satisfy you.
Yours sincerely,

Magali Deschamps-Cottin

Reviewer 2 Report

The manuscript has substantially improved in the sense, that all my remarks were dealt with, the English is also much better. Still, I hesitate a bit, whether the division of analyses by individual months is necessary, or not. I leave it at the discretion of the authors, but still recommend them to go through the manuscript once more, to reconsider the separate months issue, and to try to condense the manuscript a bit, without scarifying its information content.

Author Response

Dear reviewer

Thank you for your comments, which have improved this article.

We have attached a revised version of the manuscript with the corrections underlined in blue.

1/Results :

  • We have simplified the paragraph that describes Table 1. lig 258-277.
  • We have also modified an element in the legend of Table 1 and corrected a typographical error.

2/Discussion :

You suggested condensing the manuscript, so we simplified a sentence and deleted a paragraph that brought nothing to the discussion. Lig 446-447 and 515-518

I hope that this new version will satisfy you.

Yours sincerely,

Magali Deschamps-Cottin
